# Physico-Mechanical Optimization and Antimicrobial Properties of the Bionanocomposite Films Containing Gallic Acid and Zinc Oxide Nanoparticles

**DOI:** 10.3390/nano13111769

**Published:** 2023-05-31

**Authors:** Azin Karami, Babak Ghanbarzadeh, Leila Abolghasemi Fakhri, Pasquale M. Falcone, Mohammadyar Hosseini

**Affiliations:** 1Department of Food Science and Technology, Faculty of Agriculture, University of Tabriz, Tabriz P.O. Box 51666-16471, Iran; 2Department of Food Engineering, Faculty of Engineering, Near East University, 99138 Nicosia, Northern Cyprus, Turkey; 3Department of Agricultural, Food and Environmental Sciences, University Polytechnical of Marche, Brecce Bianche 10, 60131 Ancona, Italy; 4Department of Food Science and Hygiene, Faculty of Para-Veterinary, Ilam University, Ilam P.O. Box 69315-516, Iran

**Keywords:** optimization, carrageenan–gelatin bionanocomposite, mechanical properties, physical properties, antimicrobial packaging

## Abstract

The mechanical and physical properties of the bionanocomposite films based on κ-carrageenan (KC)–gelatin (Ge) containing zinc oxide nanoparticles (ZnONPs) and gallic acid (GA) were optimized using the response surface method, and the optimum amounts of 11.19 wt% GA and 1.20 wt% ZnONPs were obtained. The results of XRD, SEM, and FT-IR tests showed the uniform distribution of the ZnONPs and GA in the film microstructure, and suitable interactions between biopolymers and these additives, which led to increasing the structural cohesion of the biopolymer matrix and improving the physical and mechanical properties of the KC–Ge-based bionanocomposite. In the films containing gallic acid and ZnONPs, an antimicrobial effect was not observed against *E. coli*; however, the GA-loaded and optimum films show an antimicrobial effect against *S. aureus*. The optimum film showed a higher inhibition effect against *S. aureus* compared to the ampicillin- and gentamicin-loaded discs.

## 1. Introduction

Environmental and biological problems caused by petroleum-based packaging materials have increased interest in the production of degradable packaging materials using biopolymers such as polysaccharides and proteins [1]. Biopolymer-based packaging films have limited application in industry due to poor mechanical, thermal, and moisture barrier properties compared to synthetic polymer-based ones. In recent years, the improvement of biopolymer-based films for usage in food packaging has attracted more attention, owing to their biodegradability, non-toxicity, renewability, and availability. The incorporation of crosslinking agents into the film matrix and the production of composite and nanocomposite films have been introduced as effective methods in this field [2]. 

Gelatin is one of the most important biopolymers with widespread applications in the food industry, and gelatin nanocomposites can be used in film production and active food packaging [3,4]. Partial degradation or thermal denaturation of collagen causes the formation of gelatin [5] Carrageenans are water-soluble sulfated polysaccharides obtained from seaweeds of the *Rhodophyceae* family. These polysaccharides consist of long, linear chains of D-galactose and D-anhydrogalactose with ester sulfates. κ-carrageenan (KC) contains one sulfate group per repeat dimer, located on the O-3 galactose ring [6]. Carrageenan-based films have been shown to have several functional properties as packaging materials for food packaging applications [7]. 

Due to their low cost, good UV absorbance properties, abundance, high chemical stability, high resistance to processing conditions such as high temperature and pressure, and effectiveness at low concentrations, ZnONPs have attracted the attention of many researchers. The presence of ZnONPs in the polymer matrix can modify their optical, antimicrobial, mechanical, thermal, barrier, and electrical properties [8,9,10]. Polyphenols, such as gallic acid (GA, 3,4,5-trihydroxybenzoic acid, C_7_H_6_O_5_), are a group of bioactive compounds that can be used in active food packaging materials due to their antioxidant and antimicrobial properties [11]. Studies have shown that GA can also act as a plasticizer and reinforcing agent in biodegradable films, and can reduce the film’s brittleness problems and increase its tensile strength and flexibility [12,13]. 

Recently, the combined effects of plasticizers and cinnamon essential oil on the mechanical and water barrier properties of semi-refined carrageenan-based edible film have been investigated [14]. In another study, an active edible film based on KC and Ge was developed to visually detect the freshness of grass carp fillet by changing the color of the film [15]. The effect of dialdehyde κ-carrageenan and thymol-loaded zein nanoparticles on the physical and antimicrobial properties of Ge film have been investigated [16]. Lower moisture content, WVP, total soluble matter, and higher tensile strength of the film containing KC, compared to the pure gelatin film, have been reported. No notable effect on the improvement of gelatin-based film properties by zein nanoparticles has been reported, but the presence of thymol has shown acceptable antioxidant and antimicrobial activities [16].

In this study, we hypothesize that the combined use of both compounds (GA and ZnONPs) could have a synergistic effect on the different properties of the KC–Ge nanocomposites. Response surface methodology (RSM) is a beneficial method for modeling and analyzing problems where the response is affected by multiple factors. This method can be applied for multivariate experimental design, statistical modeling, and optimization of response by assessing both the interaction and the individual effects of various independent variables under fewer trial runs. It also allows researchers to evaluate the effects of multiple factors and their interactions on the dependent variables. Therefore, it is a beneficial procedure to reduce time, cost, and the number of trial runs [3,9]. There are several studies in which the central composite design (CCD) using RSM has been used effectively in optimizing film compositions to achieve the desired composite film properties [2,17,18].

In recent years, some researchers have shown the synergistic effect of NPs with each other and with other improver compounds in polymers matrices [19,20]. However, the synergistic effects of polyphenols and nanofiller incorporation in the polymer have been scarcely studied [16]. To our knowledge, although various efforts have been made to improve the basic properties of films, studies on the development of protein–polysaccharide-based active packaging films are still limited, and no previous study has used ZnONPs and polyphenols together to evaluate probable synergistic effects on the physico-mechanical and antimicrobial properties of the KC–Ge film. Therefore, in the present research, the aim was to study and optimize the interaction effects of GA polyphenol and ZnONPs as active and reinforcing agents on the important physico-mechanical and antimicrobial properties of the KC–Ge-based active nanocomposite films. A well-dispersed microstructure of nanocomposites is highly desirable for achieving good barrier, color, and mechanical properties. Therefore, the morphology and structural characteristics of the resulting nanocomposite films and the distribution quality of these additives in the film matrix were investigated by X-ray diffraction (XRD) and scanning electron microscopy (SEM) methods. In addition, an FT-IR test was conducted to confirm the possible interactions between GA, ZnONPs, and the biopolymer matrix. The antimicrobial activity of the films was assessed against *Staphylococcus aureus* (Gram^+^) and *Escherichia coli* (Gram^−^).

## 2. Material and Methods

### 2.1. Material

κ-carrageenan was purchased from an Iranian dairy industry company, Pegah, Urmia, Iran. Bovine gelatin with bloom 230–250 g was purchased from Sahand Company, Tabriz, Iran. Zinc oxide nanoparticles powder (rod shape, 10–30 nm, and purity of >99%) was purchased from US Research Nanomaterials Inc., Houston, Texas, USA. Gallic acid (molecular weight of 170.12 g mol^−1^ and a purity of 97%( was purchased from Sigma Aldrich. *Staphylococcus aureus* (ACTT 25923), *Escherichia coli* (ATCC 25922), gentamicin (10 µgr), and ampicillin antibiotic discs (10 µgr) were obtained from the culture collection at the Department of Microbiology Laboratory of Sina Hospital, Tabriz, Iran. Mueller–Hinton agar and nutrient agar culture mediums were purchased from Merck, Germany. Other chemicals were purchased from Sigma Aldrich (Darmstadt, Germany).

### 2.2. Preparation of Composite and Nanocomposite Films

The nanocomposite films were prepared by the solution casting method [21]. Different amounts of GA and ZnONPs were separately dispersed in distilled water at 30 °C for 75 min under magnetic stirring. The ZnONPs dispersion was then treated with ultrasonic prop (50 W, 50 rpm) for 20 min. A KC film-forming solution was prepared by dispersing KC powder in distilled water (1.75 wt%) at 90 °C for thirty min under stirring. Gelatin was separately dissolved in distilled water (0.5%*w/w*) at 60 °C, and then this solution was added to the KC solution at 60 °C and stirred for 40 min under magnetic stirring. The ZnONPs dispersion was added to the biopolymer solution at 45 °C and stirred for 30 min, and GA solution was then added and mixed thoroughly at 45 °C for 1 h. Finally, glycerol was added as a plasticizer (62%*w/w* biopolymer). The resulting solution was gently mixed at 45 °C for 15 min and was treated with an ultrasonic bath (40 kHz, 10 min) to remove air bubbles. Then, 50 mL of the film-forming solution was cast on a petri dish (10 cm) and dried at room temperature for 48–72 h. The thickness of the resulting films was 0.15 ± 0.05 mm. A film sample without any ZnONPs and GA added was used as a control.

### 2.3. Characterization of the Films

#### 2.3.1. Water Vapor Permeability (WVP)

The WVP of the films was determined using an ASTM standard method [22] with some modifications [9,23]. Each film (conditioned at RH = 55 ± 3%, 25 °C for 24 h) was placed on the glass vial (mouth diameter = 2 cm, depth = 4.5 cm) containing anhydrous calcium sulfate (3 g, RH = 0%). Then, all vials were weighed and transferred to a desiccator including saturated potassium sulfate (97 ± 2% RH, 25 °C), and the water vapor transferred through the film was measured by weighing the vial at different times until a stationary state was reached. The slope of the weight changes of vial vs. time was calculated by linear regression. *WVP* (g/m.s.Pa) was determined using Equation (1):(1)$$WVP=\frac{WVTR\times X}{P\times \left(R_{2}-R_{1}\right)}$$
where the water vapor transmission rate (*WVTR*) was determined by dividing the slope of the linear portion of the curve obtained for each vial by the total film surface area (m^2^). *P* is the pure water vapor pressure (Pa) at the test temperature (25 °C), *R*_2_ and *R*_1_ are the %RH in the desiccator and the vial, respectively, and *X* is the film thickness (m). All measurements were performed in 3 replicates.

#### 2.3.2. Color Measurement

The *b** (yellow-blue), lightness (*L**), and *a** (red-green) color parameters were determined by a colorimeter using a white standard plate (*b* = 0, *a* = 0, and *L** = 65). Three replications were conducted for each trial. The color difference (Δ*E*) and yellowness index (*YI*) were measured as
(2)YI=142.86 b/l
(3)$$\Delta E={\left({\left(L_{standard}-L_{sample}\right)}^{2}+{\left(a_{standard}-a_{sample}\right)}^{2}\;+{\left(b_{standard}-b_{sample}\right)}^{2}\right)}^{0.5}$$

#### 2.3.3. Mechanical Properties

Ultimate tensile strength (*UTS*), maximum strain (*S_max_*), and Young’s modulus (*YM*) were measured as indicators of mechanical properties of films using dumbbell specimens according to an ASTM standard test method D882-91 [24] on a tensile test machine (Sanaf Universal Testing Machine, Tehran, Iran). The dumbbell shaped tensile test samples with an arc transition and a maximized fillet radius are a better choice because such geometry lowers stress concentrations [25]. Samples were conditioned in a desiccator containing saturated calcium-nitrate solution (RH = 55%, for 24 h). The initial distance between the two grips was 50 mm and the stretching rate was set at 5 mm/min. Each of the mechanical properties was measured in three replications and the mean value was reported.

#### 2.3.4. X-ray Diffraction (XRD)

The distribution of ZnO nanoparticles in the film matrix and the degree of crystallinity were studied using X-ray diffraction (Siemens D5000 X-ray, Munich, Germany), with CuKα radiation (λ = 0.154 nm) in the diffraction angles range of 10° < 2θ < 80° at 40 kV and 40 mA. The scan rate was 10°/min and the step size was 0.05.

#### 2.3.5. Fourier Transform Infrared (FT-IR) Spectrometry

The tensor 27 FT-IR spectrophotometer (Bruker, Munich, Germany) was used for recording the infrared spectrum of the specimens, at wave numbers in the range of 4000–400 cm^−1^ with the resolution of 4 cm^−1^ for 100 scans, according to the method described by Perez-Mateos et al. [26].

#### 2.3.6. Scanning Electron Microscopy (SEM)

The morphology of the films was examined by scanning electron microscopy (SEM) (MIRA3 FEG-SEM, Tescan, Brno, Czech) (acceleration voltage = 15 kV). To prevent charging under the electron beam, the surface of samples was first coated with a layer of gold before SEM imaging.

#### 2.3.7. Antimicrobial Properties

The antimicrobial properties of films were assessed using the disk diffusion method as described, with some modifications [27]. The two types of bacteria (*S. aureus* as an indicator of Gram-positive bacteria and *E. coli* as an indicator of Gram-negative bacteria) were cultured in nutrient agar for 24 h at 37 °C. The cultured bacteria were inoculated into saline water until the concentration of bacteria were 0.5 McFarland scale. Then, 100 µL of the bacterial suspensions (0.5 McFarland) were spread on Mueller–Hinton agar plates. Two circular film discs (6 mm diameter, sterilized under UV light) were placed on each plate that contained bacterial culture. Two antibiotic discs including gentamicin and ampicillin were also placed on the separate plates. After incubation for 24 h at 37 °C, the diameter of inhibition zones (mm) was determined. The average values of the diameters of the inhibition zones of two films and antibiotics with two discs each were reported. The experiments were performed in triplicate.

### 2.4. Statistical Analysis and Experimental Design

The relationship between the responses or dependent variables (*WVP*, Δ*E*, *YI*, *L*, *UTS*, *S_max_*, and *YM* of films) and GA concentration (*X*_2_, %*w/w*) (15.04, 6.48, 10.03, 13.57, 5.01) and ZnONPs concentration (*X*_1_, %*w/w*) (2.00, 1.71, 1.00, 0.30, 0.00) as independent variables was investigated by RSM using a central composite design with thirteen trial runs and five repetitive central points. Appropriate levels of independent variables were determined using prior research, and the optimal level was reported in those studies [8,28]. All tests were performed in three replications, and quadratic polynomial equations were fitted to the responses (Equation (4)):(4)$$Y=\beta_{0}+\overset{k}{\underset{i=1}{\sum }}\beta_{i}X_{i}+\overset{k}{\underset{i=1}{\sum }}\beta_{ii}X_{i}^{2}+\overset{k-1}{\underset{\begin{array}{c} i=1\\ i<j \end{array}}{\sum }}\overset{k}{\underset{j=2}{\sum }}\beta_{ij}X_{i}X_{j}$$
where *Y* is a response; *X_i_* and *X_j_* are independent variables; *β_i_*, *β_ii_*, *β_ij_*, *k*, and *β*_0_ are linear, quadratic, and interaction terms, the number of studied factors, and a constant coefficient, respectively.

Design expert version 10.0.0 software was used to analyze the data and plot the response surface diagrams.

Statistically significant differences between means were calculated by the one-way analysis of variance (ANOVA) and Duncan’s multiple range test at a 0.05 significance level using SPSS software (Version 22, SPSS Inc., Chicago, IL, USA).

### 2.5. Overall Optimization of the Variables

The numerical optimization method was used to obtain the optimal values of *X_1_* and *X_2_* and the best nanocomposite film formulation in terms of physical properties.

### 2.6. Verification Experiments and Validation of the Model Equations

Verification tests using the optimal levels of additives were used to confirm the adequacy of the equations obtained. To determine the validation of the regression models, the experimental data were compared with the predicted data. The following equation was used to obtain the percentage error:(5)$$Error\;\left(\%\right)=\frac{R_{t}-R_{p}}{R_{t}}\times 100$$

In this equation, *R_t_* is the actual data obtained during the validation experiments, and *R_p_* is the predicted data obtained by the software.

## 3. Results and Discussion

### 3.1. Optimization of ĸ-Carrageenan–Gelatin-Based Nanocomposites

#### 3.1.1. Analysis of Regression Model

The experimental design, the independent variables, and the responses obtained are shown in Table 1. The fitting of the various models (linear, interactive, and quadratic) to experimental data was carried out to obtain the regression equations. The sequential model sum of squares, model summary statistic, and lack-of-fit tests were used to evaluate the model’s adequacy (Appendix A).

The sequential model sum of squares (SMSS) results for *L*, *YI*, and Δ*E* indicated a significant *p*-value for the quadratic model (*p* ≤ 0.01) (Appendix A). All three models had a *p*-value less than 0.05 for *WVP* and *S_max_*_._ However, for *UTS* and *YM* responses, the *p*-value was significant only for the linear model (*p* < 0.01). For *YM*, all three models—for ∆*E*, *YI*, and *L*, the quadratic model, for *WVP* and *S_max_* the interactive and quadratic model, and for *UTS*, the linear model—indicated a non-significance lack of fit (Appendix A). According to the above results and the results of the model summary statistic (Appendix A), in general, the quadratic model with the highest determination coefficient (R^2^), adj-R^2^ (adjusted R-Squared), and pre-R^2^ (predicted R-Squared), as well as minimum PRESS and Std. Dev. was selected as the most appropriate model for *YI*, *L*, Δ*E*, *S_max_*, and *WVP* responses. Additionally, the linear and interactive models were identified as the most suitable models for *UTS* and *YM* responses, respectively.

ANOVA and regression analysis have been used to analyze the fitness and adequacy of the selected models (Table 2). Models are well fit to the data if they have a significant and non-significant regression model and lack of fit, respectively. The *p*-values and F-test were used to evaluate the significance of the models and their coefficients [29]. As indicated in Table 2, the significant model (*p* < 0.001) and no significant lack of fit (*p* > 0.05) showed the adequacy of the models developed for all dependent variables.

The determination coefficient value can be considered as a measure for describing the extent of variations in responses by the model. This value obtained for the *L*, *YI*, Δ*E*, *WVP*, *UTS*, *S_max_*, and *YM* responses was 0.9899, 0.9872, 0.9892, 0.9674, 0.9058, 0.9721, and 0.8797, respectively. High R^2^ values (>80%) suggest that all models have a good fit and can explain the dependent variables well [30]. For all responses, the values of pre-R^2^ were in reasonable agreement with adj-R^2^, which indicated the accuracy and general availability of the polynomial models. The coefficient of variation (C.V.) is a measure that expresses the Std. Dev. vs. a percentage of the mean, and low amounts of it give better reproducibility. In this study, all responses except *WVP* and *YM* have the C.V. as less than ten percent. Given that C.V. should not exceed 10%, the C.V. values for responses indicated a good deal of reliability and a high degree of precision of the results obtained [31]. The adequate precision, which determines the signal-to-noise ratio, was higher than four for all responses, which demonstrates an adequate signal (Table 2). Thus, the created models were considered reasonable for analyzing the dependent variables.

The results of Table 2 show that the linear and quadratic effects of zinc oxide nanoparticles and gallic acid concentrations are the more significant parameters affecting Δ*E* and *YI* responses. However, For *L**, the linear effect of both additives and the quadratic effect of GA were the more significant parameters. Regarding *WVP*, the linear effect of both additives and for *UTS*, *S_max_*_,_ and *YM*, the linear effect of ZnO nanoparticles were the more significant parameters. Considering the regression coefficients, the most important factor influencing the *YM*, *WVP*, *S_max_*, *UTS*, and *L** responses was the linear effect of the ZnO concentration (*X_1_*), while the Δ*E* and *YI* responses were most affected by the linear effect of gallic acid (*X_2_*). Finally, by eliminating non-significant factors, polynomial equations were expressed using coded levels of factors to explain the effects of ZnO and GA on the responses as follows:(6)$$Y_{1}=46.36+11.11X_{1}-12.87X_{2}+4.20X_{1}X_{2}+7.92X_{1}^{2}+6.97X_{2}^{2}$$
(7)$$Y_{2}=55.45-4.18X_{1}+4.05X_{2}+1.23X_{1}X_{2}-2.73X_{1}^{2}-2.36X_{2}^{2}$$
(8)$$Y_{3}=20.39+4.02X_{1}-4.64X_{2}-1.11X_{1}X_{2}+2.88X_{1}^{2}+2.48X_{2}^{2}$$
(9)$$Y_{4}=6.603\times 10^{-11}-(3.318\times 10^{-11})X_{1}+\left(2.053\times 10^{-11}\right)X_{2}-\left(1.841\times 10^{-11}\right)X_{1}X_{2}+\left(1.574\times 10^{-11}\right)X_{2}^{2}$$
(10)$$Y_{5}=4.74+1.36X_{1}$$
(11)$$Y_{6}=28.76-6.97X_{1}-2.44X_{2}+1.62X_{1}^{2}-4.12X_{1}X_{2}$$
(12)$$Y_{7}=62.51+31.88X_{1}+11.68X_{2}$$
where *Y*_1_–*Y*_7_ are *YI*, *L**, Δ*E*, *WVP*, *UTS*, *S_max_*_,_ and *YM*, respectively.

Therefore, in short, different models were fitted to experimental data, regression equations were obtained, the adequacy of the developed models and equations were confirmed for all dependent variables, and the significant parameters and the most important factors affecting the responses were determined.

#### 3.1.2. Analysis of Response Surface

##### The Combined Effect of ZnONPs and GA on the Water Vapor Permeability (WVP)

The GA and ZnONPs concentration indicated the interaction and quadratic effect on the water vapor permeability of films (Figure 1). At all levels of GA, with increasing ZnONPs, the *WVP* value decreased, and at low ZnONPs levels, with increasing gallic acid, the *WVP* value increased. Low *WVP* was obtained in samples containing high levels of ZnONPs and intermediate levels of gallic acid.

The diffusivity and solubility of molecules of water in the hydrophilic matrices were determined using the water vapor transmission through a biopolymer [32]. The reduction in *WVP* of the KC–Ge film containing both additives can be due to (1) formation of a tortuous pathway for water vapor molecules to pass through, (2) reduction in free spaces between polymer chains, and (3) reduction in the solubility of water vapor molecules in the biopolymer matrix due to the reduction of free –OH groups in it caused by hydrogen interactions between the added components (ZnONPs and GA) and the matrix [33,34]. All of these factors reduce the solubility, diffusivity, and diffusion rate of water vapor molecules and thus reduce the water vapor permeability of the film. It should be noted that with low amounts of ZnONPs, *WVP* increased with GA content, and the reducing effects on *WVP*, described by the mentioned mechanisms, appeared with high amounts of ZnONPs and low and medium amounts of GA. The NPs dispersion quality is probably improved, and the additives–biopolymers interactions are probably increased in the presence of GA.

Similar results were obtained on the effect of zinc oxide nanoparticles [33,35] and gallic acid [34] on the *WVP* of carrageenan films.

##### The Combined Effects of ZnONPs and GA on Color Properties

The transparency and color of packaging materials are two important parameters in terms of consumer acceptance. The interaction effects of variables on Δ*E*, *YI*, and *L* were assessed using the 3D response surface plots. The variables showed the interaction and quadratic effect on all the color parameters studied (Figure 1). At low amounts of ZnONPs and high amounts of GA, the lightness index showed a maximum value (Figure 1) and vice versa. At all concentrations of ZnONPs, the *L** value increased with increasing GA. GA may have been able to increase the lightness of the film by improving the dispersion quality of ZnONPs in the matrix. The significant reduction in this parameter at high and low concentrations of NPs and GA, respectively, may be attributed to the agglomeration of NPs. The opposite trend was observed in the *YI* index and Δ*E* responses when changing the concentrations of the additives (Figure 1). As mentioned, GA probably reduced these two parameters by improving the dispersion quality of NPs.

##### The Combined Effect of ZnONPs and GA on Mechanical Properties

ZnONPs indicated a linear effect on *UTS*, and GA did not show a significant effect on this property (*p* < 0.05) (Figure 1). At all concentrations of GA, the *UTS* increased with an increase in the concentration of ZnONPs. The highest value of *UTS* (7.52 MPa) was observed in the film containing the highest amount of ZnONPs.

In general, the increase in *UTS* of the films by ZnONPs can be attributed to [8] (1) the crystalline structure of ZnONPs and the high tensile strength of these nanoparticles, (2) the interactions between nanoparticles and the polymer matrix, (3) the increase in crystalline regions in the matrix due to the addition of ZnO nanoparticles.

Researchers have obtained similar results on the effect of zinc oxide nanoparticles on the *UTS* of carrageenan films [36,37].

ZnONPs and GA showed the interaction and quadratic effects on maximum strain (*S_max_*) (Figure 1). The two variables had a synergistic effect on *S_max_*, and at high concentrations of ZnONPs, the *S_max_* decreased with the increasing GA concentration. The opposite trend was obtained for the *S_max_* with increasing GA content at low ZnONPs content. At all levels of GA, the *S_max_* decreased with increasing ZnONPs. At high concentrations of both additives, ZnONPs may have provided a high reinforcing effect through high interactions with both polymer matrix and GA, resulting in increasing the *UTS* and decreasing the *S_max_* of the film. In a similar study, Meindrawan et al. (2016) showed a significant increase in *UTS* and a non-significant decrease in elongation at the break point of the carrageenan–beeswax film due to the addition of ZnONPs [33]

The variables demonstrated a linear effect on the *YM* (*p* < 0.05) (Figure 1). Increasing both variables increased *YM* significantly (*p* < 0.05), and *YM* showed high values at high concentrations of ZnO and GA. According to the results, the two variables had a synergistic effect on *YM*. Materials reinforced with stiff fillers show higher values of *YM* because the modulus of the composite depends on the ratio of the modules of its constituent phases. Zinc oxide nanoparticles can act as a filler in the biopolymer matrix and show different effects depending on the dispersion quality of the nanoparticles within the film matrix, and the interfacial interactions between the filler and the matrix. In general, high interfacial interactions between the two additives and the polymer matrix probably improved the mechanical properties of the biopolymer by facilitating stress transfer to the reinforcement phase. The increase in interactions is caused by the extensive surface area because of the high aspect ratio, and the nanoscale dimensions of the nanofiller phase which is uniformly dispersed in the polymer matrix. In this study, the use of ZnONPs in combination with GA probably led to better dispersion of ZnONPs in the polymer matrix, which resulted in an increased aspect ratio of these particles and improved mechanical parameters of the film.

Thus, in summary, the sample containing the highest amount of ZnONPs and medium level of GA showed the lowest *WVP*. At the lowest and highest amount of ZnONPs and GA, respectively, the brightest film with the least yellowness was obtained. The film containing the highest amount of ZnONPs showed maximum *UTS*, and the two additives had synergistic effects on *S_max_* and *YM*.

Our findings showed that higher physical and mechanical properties of the film are obtained by combining gallic acid and zinc oxide nanoparticles at different levels. This is why the response surface optimization method is well-suited for this study to achieve maximum efficiency in the improvement of physico-mechanical properties. As observed, RSM provides an opportunity to simultaneously investigate changes in these properties as a function of independent variables with a minimum number of experimental points, demonstrating the convenience of the RSM to be applied in this study. Furthermore, it was also possible to evaluate the changes in the physico-mechanical properties at any value of response factors included between the lower and upper limits by the response surface methodology. That is why this method is used as one of the most popular methods for optimization purposes.

#### 3.1.3. Overall Optimization of the Variables

Numerical optimization was performed to obtain the optimal concentration of ZnONPs (*X_1_*) and GA (*X_2_*) for achieving the optimal responses, and producing the KC–Ge- based nanocomposites with the best physical and mechanical properties. For this purpose, the desired target for each response and factor was set to “within the range”, “maximum”, or “minimum”, and given each response’s importance and the study aim, a value of importance was selected for each response (Appendix A). The optimization results showed that the optimum nanocomposite composition contained 1.20% (%*w/w*) ZnONPs and 11.19% (%*w/w*) GA.

#### 3.1.4. Verification Experiments and Validation of the Model Equations

The experimental and predicted data of dependent variables at the optimum point (1.20% (%*w/w*) ZnONPs and 11.19% (%*w/w*) GA) are presented in Table 3. Only a small percentage error was observed between the true and estimated data. These true and estimated data were reasonably close to each other. Thus, acceptable percentage error (<30%) [38] indicated the validity and adequacy of the response surface models.

### 3.2. X-ray Diffraction (XRD)

Physico-chemical properties of nanocomposites depend on the uniformity of their structure and the distribution quality of NPs in the polymer matrix. Uniform distribution and non-agglomeration of NPs lead to a homogeneous and uniform film structure, greater interactions between NPs and the biopolymer chains, and improved film properties. Figure 2 shows the X-ray diffraction pattern of pure ZnONPs and GA powder, control film, control + GA (11.19%*w/w*), control + ZnONPs (1.20%*w/w*), and optimum film (1.20%*w/w* ZnONPs and 11.19%*w/w* GA). Peaks at 2θ= 32°, 2θ= 34°, 2θ= 38°, 2θ=49°, 2θ= 57°, 2θ= 63° 2θ= 66°, 2θ= 68°, 2θ= 70°, and 2θ= 72° in the diffractogram of ZnONPs powder are assigned to the (100), (002), (101), (102), (110), (103), (113), (200), (201), and (202) crystallographic planes of ZnO, respectively [39,40,41]. The control film indicated a broad peak at 2θ = 18–28° (Figure 2), which indicated the semi-crystalline structure of the biopolymer matrix. A comparison of the XRD pattern of the control and control + ZnO films showed that the addition of ZnONPs did not affect the film structure. The reduction in the peak intensity of ZnONPs in nanocomposites (Figure 2) was attributed to the low concentration, uniform distribution, and non-agglomeration of these NPs in the biopolymer matrix. This reduction can be due to the ultrasound treatment during the film preparation process [42].

Pure GA powder showed sharp peaks at 2θ= 16°, 19°, 24°, 25°, 31°, 33°, 41°, and 43°, indicating the crystalline structure of GA. The reduction in these peaks’ intensity in the films containing this compound can be ascribed to the uniform dispersion of GA in the matrix. No new peak was observed in the diffractogram of the optimum film, and only the intensity of the peaks was changed. The peaks at 2θ = 31–40° in the XRD pattern of the optimum film were attributed to the ZnONPs. The reduction in the peak intensity of ZnONPs, and the disappearance of the GA peaks in the optimum film diffractogram were probably due to the more uniform distribution and non-agglomeration of these two additives in the optimum film matrix compared to the films containing one of these compounds. These findings were consistent with the X-ray diffraction results of carrageenan–ZnONPs films obtained by Roy and Rim [36]. Thus, the results generally indicated the uniform distribution of the ZnONPs and GA in the film, and the improvement in the dispersion quality of NPs in the presence of GA.

### 3.3. Fourier Transform Infrared (FT-IR) Spectrometry

The FT-IR technique was used to identify the structure of the films as a result of interactions between different molecules of KC, Ge, GA, and ZnONPs. Figure 3 shows FT-IR spectra of GA, ZnONPs, control film, and films containing GA, ZnONPs, as well as two additives together at the optimum concentrations. Similar bands were observed at 3430, 1400, 885, and 659 cm^−1^ in ZnONPs powder, control + ZnONPs, and control + ZnONPs + GA films spectra, but a slight shift in band position was observed in the spectrum of the films. There were no noteworthy changes in the spectrums, indicating that the KC–Ge structure did not change after NPs were added to the matrix. The changes in the intensity of bands of the nanocomposites would be probably due to the Van der Waals interaction between NPs and KC–Ge. In addition, changes in the band positions at 2846, 1520, and 1122 cm^−1^ of control film with the addition of NPs confirmed the interactions between ZnONPs and the biopolymer matrix [10,35].

The specific chemical groups present in the GA were characterized (Figure 3). The main characteristics of GA spectra (Figure 3) were as follows: O–H stretching at 3460 to 3100 cm^−1^, C–O stretching at 1025 cm^−1^, C=O (carboxylic acid) stretching at 1705 cm^−1^, and C–C aromatic stretching at 1618, 1538, and 1444 cm^−1^ [43,44]. Control + GA and control + ZnONPs + GA films were compared with the control film to highlight the chemical interactions between the KC–Ge matrix and GA in the films (Figure 3). The bands between 3460 and 3329 cm^−1^, which were attributed to O–H stretching vibration formed by the hydroxyl groups of KC–Ge and water, showed higher intensities for control + GA and control + ZnONPs + GA films compared to the control film (Figure 3). The increase in the intensity of the bands at 2846 and 3460–3329 cm^−1^ of the control + GA and control + ZnONPs + GA films compared to the control film confirmed the interactions between GA and the biopolymer matrix (Figure 3).

A broad band at around 3442 cm^−1^ (3460 to 3329 cm^−1^) in the FT-IR spectrum of KC–Ge film was attributed to a stretching vibration of O–H that was affected by intramolecular or intermolecular hydrogen bonds. Introducing the GA and ZnONPs into the KC–Ge matrix caused a shift in O–H stretching toward the lower wavenumber (3429, 3435, 3436 cm^−1^, in the control + ZnONPs, control + GA, and control + ZnONPs + GA, respectively). This indicated that the hydrogen bonds of the molecules in control + GA, control + ZnONPs, and control + ZnONPs + GA films were weaker compared to the control film. This is probably because of the intermolecular interaction between the –OH and $-{\mathrm{COO}}^{-}$ groups of GA and ZnONPs and the –OH groups of biopolymers. Amide A at 3460–3145 cm^−1^ is the band typically associated with the amine group NH as well as the hydroxyl group OH, both of which are associated with hydrogen bonding. The band intensity of the films containing ZnONPs and GA, and control + ZnONPs + GA film at 3460–3145 cm^−1^ was higher compared to the control film, which was attributed to N–H and O–H stretching vibrations and increased hydrogen interactions between NPs and GA with the biopolymer matrix [45]. The bands that appeared at 2908 and 2846 cm^−1^ in the control film were assigned to the C–H stretching vibrations of alkane groups in the biopolymers’ chains, and the peak at 1650 cm^−1^ was attributed to amide I. In the films containing the additives, the intensity and position of these bands were changed slightly. Amide I indicates C=O stretching coupled with CCN deformation, in-plane NH bending, and CN stretching modes. The characteristic absorption spectrum of the protein that can reflect the secondary structure of the protein is the amide I band. The amide I peak in the control + ZnONPs, control + GA, and control + ZnONPs + GA film was shifted from 1650 to 1645 cm^−1^ (Figure 3), indicating that the incorporation of ZnONPs and GA into the matrix causes conformational changes in the chains of gelatin polypeptide that lead to a change in the presence of random coils, single-helices, and disordered structures. It is also evidence of the alteration of C–O and C=O symmetric and asymmetric vibrations, likely because of the disruption of intermolecular hydrogen bonds between $-{\mathrm{COO}}^{-}$ groups resulting from incorporated GA. Amide II arises from stretching vibrations of C–N groups and bending vibrations of N–H groups. This band in control + ZnONPs, control + GA, and control + ZnONPs + GA film (Figure 3) was shifted from 1520 cm^−1^ to 1529, 1529, and 1531 cm^−1^, respectively, in comparison with control film. For KC–Ge film containing both additives, this band appeared at a significantly higher wavenumber compared to control and control + ZnONPs and control + GA films. The observed differences may be due to the change in the secondary structure of the gelatin polypeptide chains caused by incorporation of the GA and ZnONPs. Amide III represents vibrations of CH_2_ groups of glycine or in-plane vibrations of N–H and C–N groups of bound amide. As shown in Figure 3, amide III bands in the control sample (1269 cm^−1^) shifted to a lower wavenumber, i.e., 1240, 1258, and 1255 cm^−1^, respectively, when ZnONPs, GA, and both of them were incorporated into the control sample. The results showed that the hydroxyl group in GA and ZnONPs and the amino groups in Ge were consumed during the mixing process. The bands observed at 1122, 1090, 1116, and 1116 cm^−1^ in the control, control + ZnONPs, control + GA, and control + ZnONPs + GA film spectra, respectively, denote the presence of sulfate ester groups in carrageenan and bands at 1028, 1058, 1033, and 1036 cm^−1^ attributed to glycosidic linkages. Furthermore, the band observed at 927 cm^−1^ was assigned to 3,6-anhydro-D-galactose and the typical bands that appeared at 856, 860, 848, 848 cm^−1^ in the control, control + ZnONPs, control + GA, and control + ZnONPs + GA film spectra, respectively, were due to galactose-4-sulfate. The FT-IR spectra of the film containing both NPs and GA were similar to the spectra of the other samples. The changes in the position and intensity of bands indicated interactions between the additives and the film matrix [10,36,46,47,48]. These results indicated the appropriate compatibility and suitable interactions between biopolymers and the additives.

### 3.4. Scanning Electron Microscopy (SEM)

To study the morphology and structure of the films, scanning electron microscopy (SEM) was applied. Cross-section FE-SEM micrographs of the control film and the films containing GA, ZnONPs, as well as two additives together at the optimum concentrations are shown in Figure 4. The cross-sectional microstructure images showed some cracks in the control and control + GA films (Figure 4a,b). The morphology of these films can be seen at a larger magnification in Figure 4e,f. Although the addition of ZnONPs to the control film reduced the cracks in its structure (Figure 4c), larger ZnO particles (or aggregates) were clearly visible in the control + ZnONPs film micrograph compared to the film containing both GA and ZnONPs (Figure 4c,d). The addition of GA along with ZnONPs in the optimum film affected the dispersion state of ZnONPs and improved the distribution quality of these NPs in the matrix (circles in Figure 4c,d), which caused a further reduction in cracks, and led to the prevention of NPs agglomeration in the film, and generally resulted in a more cohesive and uniform film structure (Figure 4d). This was evidence of the improved dispersion quality of nanoparticles in the presence of GA, and confirmation of the effect of ZnONPs and GA in improving the dispersion quality of each other in the KC–Ge film matrix. The reduction in cracks in the optimal film can be attributed to the increased interactions between the additives and the biopolymers matrix as a result of the better dispersion quality of the additives, which was confirmed by the FT-IR results. In summary, it can be said that the simultaneous addition of additives in the film caused a good distribution of nanoparticles and prevented their agglomeration.

### 3.5. Antimicrobial Properties

Antimicrobial activity of the different films, including control film, gallic acid-loaded film, ZnONPs-loaded film, and optimum film (containing 11.19 wt% GA and 1.20 wt% ZnONPs) against *S. aureus* (Gram positive) and *E. coli* (Gram negative) pathogenic bacteria has been illustrated in Figure 5. As shown, the inhibition zone around the discs was not observed in the control film, indicating the lack of antimicrobial activity of it against these bacteria. Similarly, no antimicrobial effect was observed against *E. coli* in the films containing gallic acid and ZnONPs. However, the GA-loaded and optimum films show an antimicrobial effect against *S. aureus.* This could be attributed to the difference in the cell wall structure between Gram^+^ bacteria and Gram^−^ bacteria. The latter have a more complex cell wall than the former, which consists of a thin layer of peptidoglycan in the vicinity of the cytoplasmic membrane, and an outer membrane consisting of lipopolysaccharides, phospholipids, and proteins. In contrast, Gram^+^ bacteria do not have an outer membrane, but have a thick cell wall mainly containing peptidoglycans outside of the cytoplasmic membrane, which makes the cells rigid and makes it difficult for antimicrobial substances to penetrate through. Some researchers have stated that the presence of the outer membrane in Gram^−^ bacteria limits the diffusion of antimicrobial components through it, while the absence of this complex membrane in Gram^+^ bacteria may contribute to the greater permeability of small antimicrobial molecules, facilitating the access to the cell membrane [49].

Additionally, the ZnONPs are probably trapped in the carrageenan–gelatin biopolymer matrix, which limits their migration from this type of film and prevents its antimicrobial effects [50]. The results indicated that the diameter of the inhibition zone of the GA-loaded and optimum films discs against *S. aureus* was significantly greater than the diameter of the inhibition zone of the ampicillin- and gentamicin-loaded discs (*p* < 0.05). The results of current research were consistent with the results of [13] who reported that the GA-loaded chitosan film showed a higher antimicrobial effect against Gram^+^ bacteria (*Bacillus subtilis* and *Listeria monocytogenes*) than Gram^−^ bacteria (*Salmonella typhimurium* and *E. coli*).

## 4. Conclusions

In this study, biodegradable κ-carrageenan–gelatin-based nanocomposite films containing ZnONPs and GA were produced by the casting method, and then the physico-mechanical characteristics of the films were optimized using the RSM method. The synergistic influence of additives on the studied properties was observed. An adequate fit of the developed models with the experimental values was exhibited using analysis of variance. The optimum levels of ZnONPs and GA were obtained as 1.20 (%*w/w*) and 11.19 (%*w/w*), respectively. A good agreement was shown between the experimental and the predicted values, and verification experiments indicated the adequacy of the response surface models for predicting the optimum response values. XRD results showed uniform distribution of nanoparticles and GA in the film, and improved the dispersion quality of nanoparticles in the film matrix in the presence of gallic acid. There was also a reduction in the cracks of the KC–Ge film in the presence of both additives. The FT-IR results confirmed the interactions between matrix biopolymers, GA, and ZnONPs. The SEM test indicated that the simultaneous presence of GA and ZnONPs in the film structure caused a good distribution of nanoparticles and led to the prevention of nanoparticle agglomeration in the film. The optimum film showed a good inhibition effect against *S. aureus*, which indicated its promising effectiveness in improving the safety of food products. This developed nanocomposite packaging material can have potential applications as an active packaging film to maintain food safety and to extend the shelf life of packaged food products. Investigating the migration potential of ZnONPs from this packaging material to food systems is another potentially important field for future studies.

## Figures and Tables

**Figure 1 nanomaterials-13-01769-f001:**
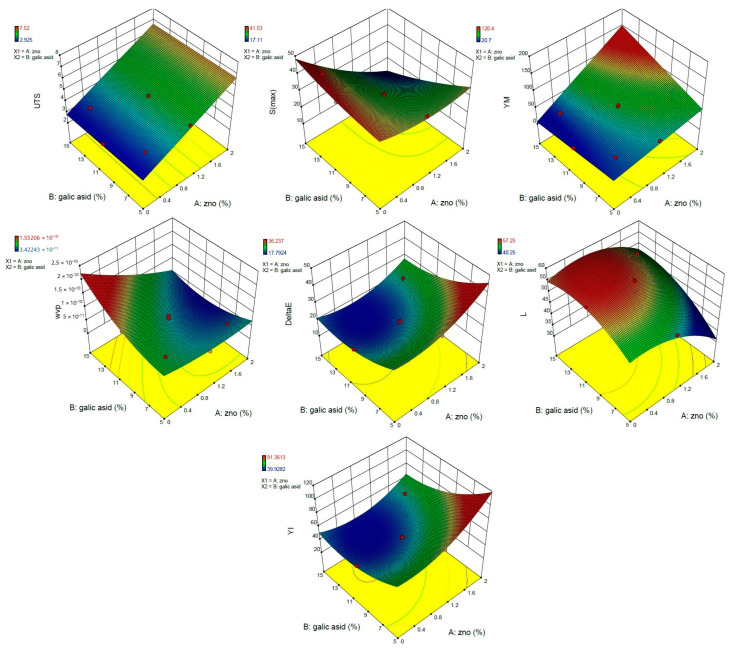
Response surfaces plots of *UTS*, *S_max_*, *YM*, *WVP*, Δ*E*, *L*, and *YI*, as function of ZnONPs and GA levels *(*in color*)*.

**Figure 2 nanomaterials-13-01769-f002:**
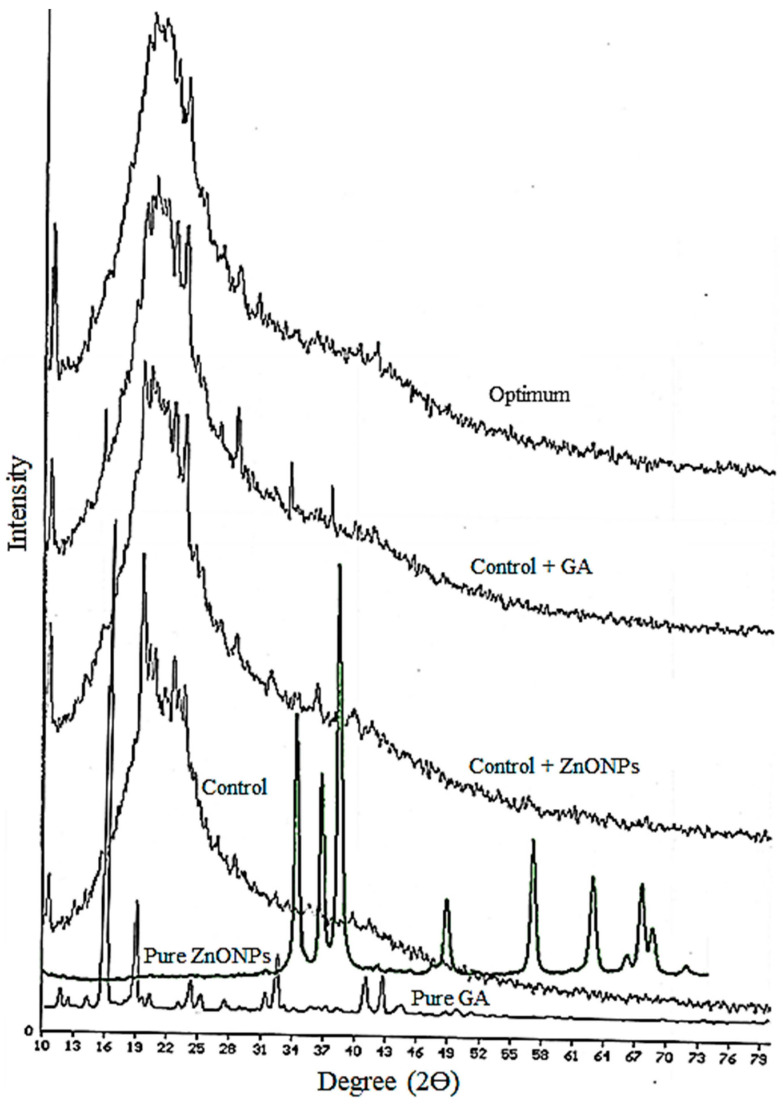
XRD diffractograms of the ZnONPs and GA powder, control film, control + GA, control + ZnONPs, and optimum film *(*black and white*)*.

**Figure 3 nanomaterials-13-01769-f003:**
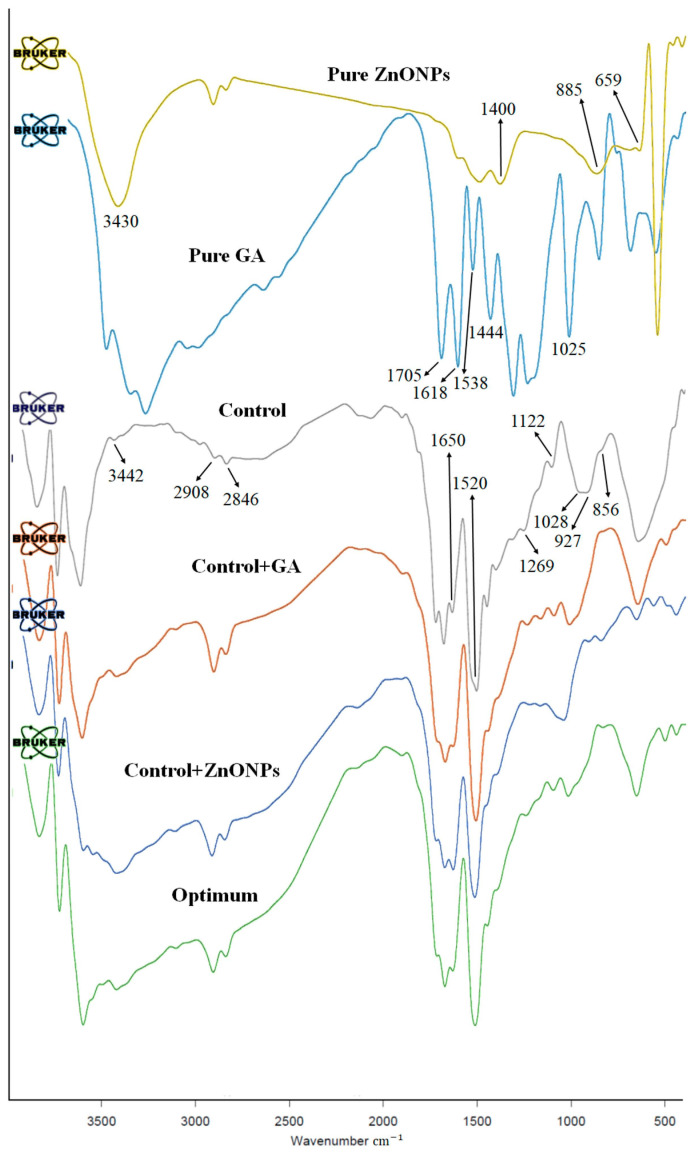
FT-IR spectrum of the ZnONPs and GA powder, control film, control + GA, control + ZnONPs, and optimum film *(*in color*)*.

**Figure 4 nanomaterials-13-01769-f004:**
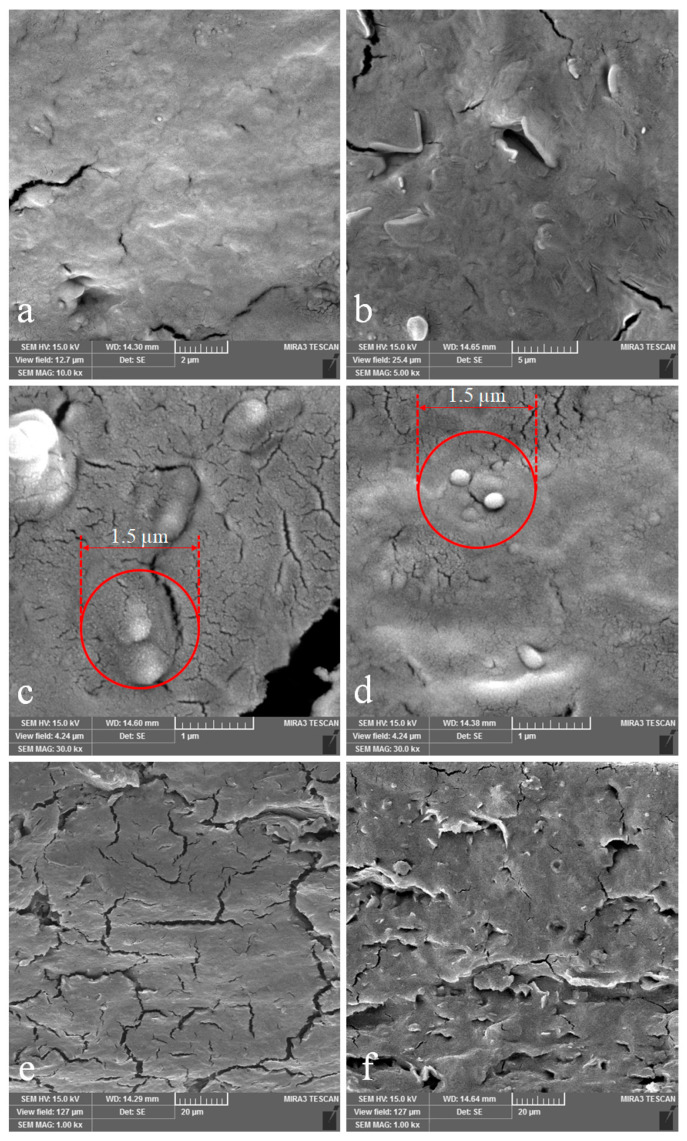
Cross-section FE-SEM micrographs of the (**a**,**e**) control film, (**b**,**f**) control + GA, (**c**) control + ZnONPs, and (**d**) optimum film *(*black and white*)*.

**Figure 5 nanomaterials-13-01769-f005:**
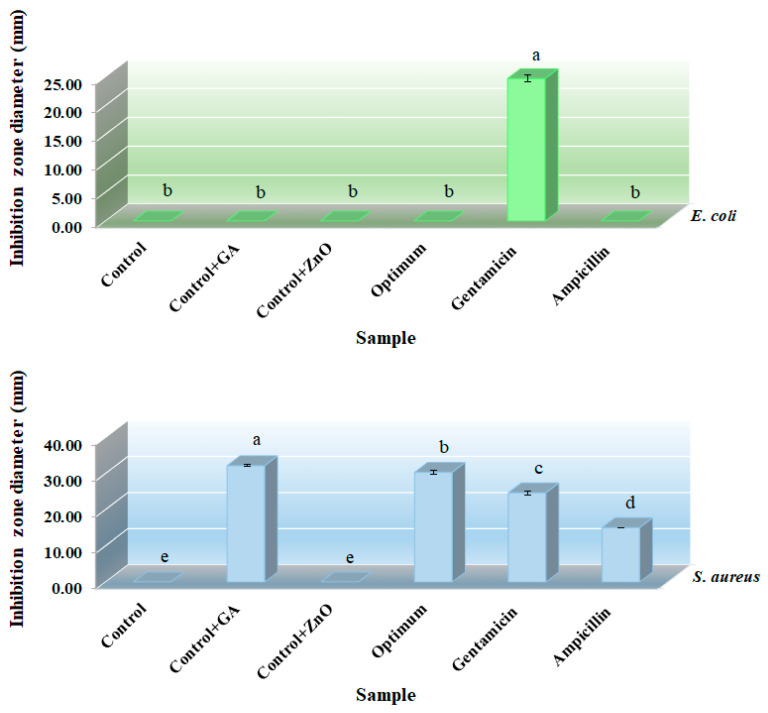
Antimicrobial activity of the control, control + GA, control + ZnONPs, and optimum films against *S. aureus* and *E. coli* (different letters (a, b, c, d, and e) show statistically significant differences (*p* ≤ 0.05)) *(*in color).

**Table 1 nanomaterials-13-01769-t001:** Coded and uncoded levels of independent variables in the central composite design and experiential data for the dependent variables.

Ruu	Coded Variables		Uncoded Variables		Responses						
	*X* _1_	*X* _2_	ZnONPs (%*w/w*)	GA (%*w/w*)	*UTS*(MPa)	*S_max_* (%)	*YM* (MPa)	*L*	*YI*	Δ*E*	*WVP* (×10^−11^ g/m·s·Pa)
1	0	0	1	10.03	4.59 ± 0.39	28.92 ± 2.53	66.05 ± 5.35	55.50 ± 0.57	42.45 ± 2.93	19.07 ± 0.87	6.71 ± 0.90
2	+1	−1	1.71	6.48	4.47 ± 0.94	29.17 ± 0.70	68.25 ± 1.49	40.25 ± 2.50	91.36 ± 2.50	36.23 ± 0.70	5.51 ± 0.81
3	−1/41,421	0	0	10.03	2.92 ± 0.97	41.53 ± 0.39	20.70 ± 1.22	56.50 ± 1.29	48.13 ± 6.33	20.82 ± 2.42	12.5 ± 0.14
4	−1	−1	0.3	6.48	3.61 ± 0.10	36.00 ± 3.00	38.50 ± 3.50	50.75 ± 5.18	58.12 ± 5.65	25.24 ± 3.01	8.67 ± 1.22
5	0	−1/41,421	1	5.01	4.99 ± 0.25	35.01 ± 2.98	52.70 ± 5.22	46.00 ± 1.41	77.69 ± 2.44	31.48 ± 0.75	6.30 ± 0.87
6	−1	+1	0.3	13.57	3.46 ± 0.41	40.43 ± 2.64	34.60 ± 4.54	57.25 ± 0.57	39.92 ± 2.07	17.79 ± 0.75	15.5 ± 0.22
7	0	0	1	10.03	4.80 ± 1.01	29.62 ± 0.28	62.55 ± 3.70	56.00 ± 0.81	47.23 ± 3.95	20.57 ± 1.50	7.74 ± 1.17
8	+1/41,421	0	2	10.03	7.52 ± 1.82	23.41 ± 3.08	120.4 ± 30.0	44.25 ± 0.81	75.89 ± 21.24	31.15 ± 6.64	3.42 ± 0.49
9	0	0	1	10.03	4.80 ± 2.31	26.22 ± 5.38	62.55 ± 6.61	55.75 ± 0.50	48.05 ± 1.48	20.91 ± 0.56	5.22 ± 0.72
10	0	+1/41,421	1	15.04	4.64 ± 1.60	26.58 ± 5.00	85.70 ± 5.50	56.25 ± 0.50	42.54 ± 1.20	18.91 ± 0.42	13.4 ± 0.12
11	0	0	1	10.03	4.30 ± 2.32	29.62 ± 3.09	34.64 ± 1.60	55.00 ± 0.81	45.48 ± 3.78	20.17 ± 1.41	6.66 ± 0.71
12	+1	+1	1.71	13.57	5.97 ± 1.59	17.11 ± 1.02	118.9 ± 8.01	51.66 ± 2.08	56.34 ± 5.72	24.35 ± 2.14	5.00 ± 0.69
13	0	0	1	10.03	4.59 ± 1.37	29.41 ± 3.98	47.10 ± 5.57	55.00 ± 1.00	48.55 ± 6.72	21.19 ± 2.44	6.66 ± 1.00

**Table 2 nanomaterials-13-01769-t002:** Regression coefficients for the responses and analysis of variance of regression models.

**Source**	**Water Vapor Permeability (g/m·s·Pa)**				** *L* **			
	**Reg. Co**	**(df)**	**F-Value**	** *p* ** **-Value Prob > F**	**Reg. Co**	**(df)**	**F-Value**	** *p* ** **-Value Prob > F**
*X* _1_	−3.318 × 10^−11^	1	119.09	<0.0001 **	−4.18	1	268.01	<0.0001 **
*X* _2_	2.053 × 10^−11^	1	45.60	0.0003 **	−4.05	1	252.28	<0.0001 **
*X_1_^2^*	6.254 × 10^−12^	1	3.68	0.0966	2.73	1	99.62	0.0625
*X* _1_ *X* _2_	−1.841 × 10^−11^	1	18.33	0.0037 **	1.23	1	11.61	0.0113
*X* _2_ ^2^	1.574 × 10^−11^	1	23.29	0.0019 **	−2.36	1	74.13	<0.0001 **
Model	-	5	41.60	<0.0001 **	-	5	137.17	<0.0001 **
Lack of fit	-	3	0.81	0.5504	-	3	4.74	0.0835
R^2^	0.9674	-	-	-	0.9899	-	-	-
R^2^-Adj	0.9442	-	-	-	0.9827	-	-	-
R^2^-Pred	0.8808	-	-	-	0.9405	-	-	-
Adeq Precision	21.299	-	-	-	33.571	-	-	-
C.V.%	10.81	-	-	-	1.38	-	-	-
Std. Dev	8.601 × 10^−012^	-	-	-	0.72	-	-	-
PRESS	1.896 × 10^−021^	-	-	-	21.47	-	-	-
**Source**	** *YI* **				**Δ** ** *E* **			
*X_1_*	11.11	1	173.72	<0.0001 **	4.02	1	208.97	<0.0001 **
*X_2_*	−12.87	1	232.89	<0.0001 **	−4.64	1	278.28	<0.0001 **
*X_1_^2^*	7.92	1	76.78	<0.0001 **	−2.88	1	93.33	<0.0001 **
*X_1_X_2_*	−4.20	1	12.43	0.0097 **	−1.11	1	7.92	0.0260
*X_2_^2^*	6.97	1	59.49	<0.0001 **	−2.48	1	69.44	<0.0001 **
Model	-	5	107.95	<0.0001 **	-	5	127.88	<0.0001**
Lack of fit	-	3	0.83	0.5415	-	3	0.78	0.5630
R^2^	0.9872	-	-	-	0.9892	-	-	-
R^2^-Adj	0.9781	-	-	-	0.9814	-	-	-
R^2^-Pred	0.9527	-	-	-	0.9609	-	-	-
Adeq Precision	29.602	-	-	-	32.409	-	-	-
C.V.%	4.30	-	-	-	3.32	-	-	-
Std. Dev	2.38	-	-	-	0.79	-	-	-
PRESS	147.14	-	-	-	15.64	-	-	-
**Source**		***UTS* (MPa)**				** *S_max_* ** **(%)**		
	**Reg. Co**	**(df)**	**F-Value**	** *p* ** **-Value Prob > F**	**Reg. Co**	**(df)**	**F-Value**	** *p* ** **-Value Prob > F**
*X_1_*	1.36	1	96.15	<0.0001 **	−6.97	1	180.57	<0.0001 **
*X_2_*	−0.016	1	0.014	0.9081	−2.44	1	22.21	0.0022 **
*X_1_^2^*	**-**	**-**	**-**	**-**	1.62	1	8.46	0.0227 *
*X_1_X_2_*	**-**	**-**	**-**	**-**	−4.12	1	31.59	0.0008 **
*X_2_^2^*	**-**	**-**	**-**	**-**	0.78	1	1.97	0.2034
Model	**-**	2	48.08	<0.0001 **	**-**	5	48.78	<0.0001 **
Lack of fit	**-**	6	5.39	0.0625	**-**	3	1.06	0.4574
R^2^	0.9058	**-**	**-**	**-**	0.9721	**-**	**-**	**-**
R^2^-Adj	0.9827	**-**	**-**	**-**	0.9522	**-**	**-**	**-**
R^2^-Pred	0.9405	**-**	**-**	**-**	0.8877	**-**	**-**	**-**
Adeq Precision	33.571	**-**	**-**	**-**	24.312	**-**	**-**	**-**
C.V.%	1.38	**-**	**-**	**-**	4.85	**-**	**-**	**-**
Std. Dev	0.39	**-**	**-**	**-**	1.47	**-**	**-**	**-**
PRESS	3.13	**-**	**-**	**-**	60.68	**-**	**-**	**-**
**Source**	***YM*** **(MPa)**							
*X_1_*	31.88	1	53.72	<0.0001 **				
*X_2_*	11.68	1	7.21	0.0250 *				
*X_1_^2^*	**-**		**-**	**-**				
*X_1_X_2_*	13.64	1	4.91	0.0538				
*X_2_^2^*	**-**		**-**	**-**				
Model	**-**	3	21.95	0.0002 **				
Lack of fit	**-**	5	0.73	0.6370				
R^2^	0.8797	**-**	**-**	**-**				
R^2^-Adj	0.8397	**-**	**-**	**-**				
R^2^-Pred	0.8069	**-**	**-**	**-**				
Adeq Precision	14.987	**-**	**-**	**-**				
C.V.%	19.68	**-**	**-**	**-**				
Std. Dev	12.30	**-**	**-**	**-**				
PRESS	2187.16	**-**	**-**	**-**				

*, **: Significant at *p* < 5% and *p* < 1%, respectively.

**Table 3 nanomaterials-13-01769-t003:** Experimental and predicted values of responses at optimal point.

**Response**	**Predicted Value**	**Experimental Value ***	**Percentage Error**
WVP (g/m.s.Pa)	6.39 × 10^−11^	6.31 × 10^−11^ ± 2.65 × 10^−12^	1.28
L	55.249	59.30 ± 1.52	6.83
ΔE	20.389	16.20 ± 1.32	25.89
YI	46.251	36.70 ± 3.26	26.01
UTS (Mpa)	5.121	4.02 ± 0.58	27.26
Smax (%)	25.829	22.45 ± 3.82	15.03
YM (Mpa)	76.554	76.82 ± 3.16	0.34

* Values are presented as mean ± SD, n = 3.

## Data Availability

The data presented in this study are available on request from the corresponding author.

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
