# Peer review of "Physico-Mechanical Optimization and Antimicrobial Properties of the Bionanocomposite Films Containing Gallic Acid and Zinc Oxide Nanoparticles"

_nanomaterials, 2023, doi:10.3390/nano13111769_

Round 1

Reviewer 1 Report

Dear Authors,

The paper concerns an interesting problem of obtaining nanocomposites. Films with changed properties can be obtained depending on the type of nanoparticles. I believe the experiments described in the publication were appropriately performed, and the discussion of the results is also correct. Nevertheless, some points require clarification.

Line 50

The Authors wrote that gelatin could be used to produce films and active food packaging. I agree with this, but I think it should be clarified that gelatin nanocomposites are proposed, not gelatin itself. Nanocomposites are also described in the articles cited in the publication (Azarifar et al., 2019; Shen et al., 2021).

Line 136

The thickness of the films obtained was 0.15 ± 0.05 mm. What was the difference in the thickness of a single foil in different places on its surface?

Lines 140-142

The authors used very small sample surfaces when testing the water vapor transmission. Why? The ASTM standard requires that the mouth of the dish shall be as large as practical and at least 3000mm2.

Lines 162 - 166

The publication uses dumbbell samples according to the standard test method. Was an extensometer used? Why were strip samples not used?

Line 510

I agree with the Authors' statement that ZnONPs can be trapped in the carrageenan-gelatin biopolymer matrix, which limits their migration from this type of film and prevents its antimicrobial activity. Has an experimental study of the migration of this substance from the nanocomposite been considered?

One of the essential parameters of packaging films is oxygen permeation. What was the reason that this parameter was not measured?

Author Response

Reviewer 1

The paper concerns an interesting problem of obtaining nanocomposites. Films with changed properties can be obtained depending on the type of nanoparticles. I believe the experiments described in the publication were appropriately performed, and the discussion of the results is also correct. Nevertheless, some points require clarification.

We appreciate the comments of the dear reviewer and his/her positive feedback. We considered these comments and we revised our manuscript according to the comments as follow:

Line 50

The Authors wrote that gelatin could be used to produce films and active food packaging. I agree with this, but I think it should be clarified that gelatin nanocomposites are proposed, not gelatin itself. Nanocomposites are also described in the articles cited in the publication (Azarifar et al., 2019; Shen et al., 2021).

Answer:

Following this comment, the change was made in the revised manuscript.

Line 136

The thickness of the films obtained was 0.15 ± 0.05 mm. What was the difference in the thickness of a single foil in different places on its surface?

Answer:

The differenc in thickness of a single film at different places on its surface was represented by the standard deviation (0.05 mm).

Lines 140-142

The authors used very small sample surfaces when testing the water vapor transmission. Why? The ASTM standard requires that the mouth of the dish shall be as large as practical and at least 3000mm2.

Answer:

Thanks for the Reviewer’s comment. WVP of the films was determined using ASTM standard method with some modifications according to the work of Fakhri et al. [1] and Ghanbarzadeh et al. [2]. We corrected in the revised manuscript.

Lines 162 - 166

The publication uses dumbbell samples according to the standard test method. Was an extensometer used? Why were strip samples not used?

Answer:

Mechanical properties were measured using a tensile test machine and the extensometer was not used. The dumbbell shaped tensile test samples with an arc transition and a maximized fillet radius are a better choice because such geometry lowers stress concentrations [3].

Line 510

I agree with the Authors' statement that ZnONPs can be trapped in the carrageenan-gelatin biopolymer matrix, which limits their migration from this type of film and prevents its antimicrobial activity. Has an experimental study of the migration of this substance from the nanocomposite been considered?

Answer:

The migration of ZnONPs from the samples was not investigated in this study, and no study was found based on the migration of these nanoparticles from the carageenan-gelatin biopolymer matrix. This statement is based on the study of Störmer et al. [4]. We added this reference in the section 3.5 and highlighted it.

One of the essential parameters of packaging films is oxygen permeation. What was the reason that this parameter was not measured?

Answer:

As dear reviewer comment, one of the essential parameters of packaging films is oxygen permeation, but unfortunately we did not have access to the oxygen permeability measurement device.

References

  1. Fakhri, L.A.; Ghanbarzadeh, B.; Dehghannya, J.; Dadashi, S. Central Composite Design Based Statistical Modeling for Optimization of Barrier and Thermal Properties of Polystyrene Based Nanocomposite Sheet for Packaging Application. Food Packaging and Shelf Life 2021, 30, 100725, doi:10.1016/j.fpsl.2021.100725.
  2. Ghanbarzadeh, B.; Almasi, H.; Entezami, A.A. Improving the Barrier and Mechanical Properties of Corn Starch-Based Edible Films: Effect of Citric Acid and Carboxymethyl Cellulose. Industrial Crops and Products 2011, 33, 229–235, doi:10.1016/j.indcrop.2010.10.016.
  3. Feng, L.; Jasiuk, I. Effect of Specimen Geometry on Tensile Strength of Cortical Bone. J. Biomed. Mater. Res. 2010, 95A, 580–587, doi:10.1002/jbm.a.32837.
  4. Störmer, A.; Bott, J.; Kemmer, D.; Franz, R. Critical Review of the Migration Potential of Nanoparticles in Food Contact Plastics. Trends in Food Science & Technology 2017, 63, 39–50, doi:10.1016/j.tifs.2017.01.011.

Reviewer 2 Report

The authors explored the optimization of bionanocomposite films based on κ-carrageenan, gelatin, zinc oxide nanoparticles, and gallic acid. The study reveals enhanced mechanical and physical properties due to uniform distribution and improved interactions between the components. While no antimicrobial effect was observed against E. Coli, the optimized films demonstrated significant antimicrobial activity against S. aureus, surpassing ampicillin and gentamicin-loaded films. The quality and potential applications of this work justify publication in Nanomaterials after addressing the following requests:

1.    Elaborate on the novelty of the study: While the authors mention some related research, it would be helpful to emphasize the unique aspects of this study.

2.    Provide more context for the response surface methodology (RSM): Briefly explain why these specific methods are well-suited for this study and how they have been successfully applied in similar research. This will help readers understand their relevance and importance to your work.

3.    The results and discussion section is well-structured, and the findings are presented clearly. It would be beneficial to highlight the main findings of each subsection in a summary format before moving on to the next subsection.

4.    The study would benefit from a more comprehensive conclusion section that summarizes the main findings and highlights their implications for the development of biopolymer-based films with improved properties for food packaging applications. The authors should also discuss any limitations of the study and potential areas for future research.

Author Response

Reviewer 2

The authors explored the optimization of bionanocomposite films based on κ-carrageenan, gelatin, zinc oxide nanoparticles, and gallic acid. The study reveals enhanced mechanical and physical properties due to uniform distribution and improved interactions between the components. While no antimicrobial effect was observed against E. Coli, the optimized films demonstrated significant antimicrobial activity against S. aureus, surpassing ampicillin and gentamicin-loaded films. The quality and potential applications of this work justify publication in Nanomaterials after addressing the following requests:

Thank you for the positive feedback and your helpful comments and suggestions on the manuscript. We considered these comments and we revised our manuscript according to the comments as follow:

  1. Elaborate on the novelty of the study: While the authors mention some related research, it would be helpful to emphasize the unique aspects of this study.

Answer:

Based on dear reviewer,s comment, we added some explanations in the introduction section about the novelty of the study and highlighted them.

  1. Provide more context for the response surface methodology (RSM): Briefly explain why these specific methods are well-suited for this study and how they have been successfully applied in similar research. This will help readers understand their relevance and importance to your work.

Answer:

According to dear reviewer suggestion, we added and highlighted more context for the response surface methodology (RSM) in the revised version at the end of the section 3.1.2.

  1. The results and discussion section is well-structured, and the findings are presented clearly. It would be beneficial to highlight the main findings of each subsection in a summary format before moving on to the next subsection.

Answer:

Thanks for the Reviewer’s comment. We added the main findings of each subsection in a summary format before moving on to the next subsection and highlighted them.

  1. The study would benefit from a more comprehensive conclusion section that summarizes the main findings and highlights their implications for the development of biopolymer-based films with improved properties for food packaging applications. The authors should also discuss any limitations of the study and potential areas for future research.

Answer:

In the continuation of this comment, we discussed some limitations of the study and potential areas for future research in the conclusions section and highlighted them.

References

  1. Fakhri, L.A.; Ghanbarzadeh, B.; Dehghannya, J.; Dadashi, S. Central Composite Design Based Statistical Modeling for Optimization of Barrier and Thermal Properties of Polystyrene Based Nanocomposite Sheet for Packaging Application. Food Packaging and Shelf Life 2021, 30, 100725, doi:10.1016/j.fpsl.2021.100725.
  2. Ghanbarzadeh, B.; Almasi, H.; Entezami, A.A. Improving the Barrier and Mechanical Properties of Corn Starch-Based Edible Films: Effect of Citric Acid and Carboxymethyl Cellulose. Industrial Crops and Products 2011, 33, 229–235, doi:10.1016/j.indcrop.2010.10.016.
  3. Feng, L.; Jasiuk, I. Effect of Specimen Geometry on Tensile Strength of Cortical Bone. J. Biomed. Mater. Res. 2010, 95A, 580–587, doi:10.1002/jbm.a.32837.
  4. Störmer, A.; Bott, J.; Kemmer, D.; Franz, R. Critical Review of the Migration Potential of Nanoparticles in Food Contact Plastics. Trends in Food Science & Technology 2017, 63, 39–50, doi:10.1016/j.tifs.2017.01.011.
